# RELATIVE MOLECULE SELF-ATTENTION TRANSFORMER

## ABSTRACT

Pretraining using self-supervised learning holds promise to revolutionize molecule property prediction – a central task to drug discovery and many more industries – by enabling data efficient learning from scarce experimental data. However, despite significant progress, non-pretrained methods can be still competitive in certain settings. We reason that architecture might be a key bottleneck. In particular, enriching the backbone architecture with domain-specific inductive biases has been key for the success of self-supervised learning in other domains. Inspired by this, we methodologically explore the design space of the self-attention mechanism for molecular data. Our main contribution is Relative Molecule Attention Transformer (R-MAT): a novel Transformer-based model that achieves state-of-the-art or very competitive results across a wide range of molecule property prediction tasks. Relative Molecule Attention Transformer uses a novel self-attention variant that can be readily incorporated into future models for processing molecular data.

## 1 INTRODUCTION

Predicting molecular properties is central to applications such as drug discovery or material design. Without accurate prediction of properties such as toxicity, a promising drug candidate is likely to fail clinical trials. Many molecular properties cannot be feasibly computed (simulated) from first principles and instead have to be extrapolated, from an often small experimental dataset (Chan et al., 2019; Bender & Cortés-Ciriano, 2021). The prevailing approach is to train a machine learning model such a random forest (Korotcov et al., 2017) or a graph neural network (Gilmer et al., 2017) from scratch to predict the desired property for a new molecule.

Machine learning is moving away from training models purely from scratch. In natural language processing (NLP), advances in large-scale pretraining (Devlin et al., 2018; Howard & Ruder, 2018) and the development of the Transformer (Vaswani et al., 2017) architecture have culminated in large gains in data efficiency across multiple tasks (Wang et al., 2019a). Instead of training models purely from scratch, the models in NLP are commonly first pretrained on a large unsupervised corpora. The chemistry domain might be at the brink of an analogous revolution, which could be especially transformative due to the high cost of obtaining large experimental datasets. In particular, recent work has proposed Molecule Attention Transformer (MAT), a Transformer-based architecture adapted to processing molecular data (Maziarka et al., 2020) and pretrained using self-supervised learning for graphs (Hu et al., 2020). Several works have shown further gains by improving network architecture or the pretraining tasks (Chithrananda et al., 2020; Fabian et al., 2020; Rong et al., 2020).

However, pretraining has not yet led to such transformative data-efficiency gains in molecular property prediction. For instance, non-pretrained models with extensive handcrafted featurization tend to achieve very competitive results (Yang et al., 2019a). We reason that architecture might be a key bottleneck. In particular, most Transformers for molecules do not encode the three dimensional structure of the molecule (Chithrananda et al., 2020; Rong et al., 2020), which is a key factor determining many molecular properties. On the other hand, performance has been significantly boosted in other domains by enriching the Transformer architecture with proper inductive biases (Dosovitskiy et al., 2021; Shaw et al., 2018; Dai et al., 2019; Ingraham et al., 2021; Huang et al., 2020; Romero & Cordonnier, 2021; Khan et al., 2021; Ke et al., 2021). Motivated by this perspective, we methodologically explore the design space of the self-attention layer, a key computational primitive of the Transformer architecture, for molecular property prediction. In particular, we explore variants of relative self-attention, which

has been shown to be effective in various domains such as protein design and NLP (Shaw et al., 2018; Ingraham et al., 2021)

Our main contribution is a new self-attention layer for molecular graphs. We tackle the aforementioned issues with Relative Molecule Attention Transformer (R-MAT), our pre-trained transformer-based model, shown in Figure 1. We propose Relative Molecule Self-Attention, a novel variant of relative self-attention, which allows us to effectively fuse distance and graph neighbourhood information (see Figure 2). Our model achieves state-of-the-art or very competitive performance across a wide range of tasks. Satisfyingly, R-MAT outperforms more specialized models without using extensive handcrafted featurization or adapting the architecture specifically to perform well on quantum prediction benchmarks. The importance of representing effectively distance and other relationships in the attention layer is evidenced by large performance gains compared to MAT.

An important inspiration behind this work was to unlock the potential of large pretrained models for the field, as they offer unique long-term benefits such as simplifying machine learning pipelines. We show that R-MAT can be trained to state-of-the-art performance with only tuning the learning rate. We also open-source weights and code as part of the Huggingmolecules (Gaiński et al., 2021) package.

## 2 RELATED WORK

Pretraining coupled with the efficient Transformer architecture unlocked state-of-the-art performance in molecule property prediction (Maziarka et al., 2020; Chithrananda et al., 2020; Fabian et al., 2020; Rong et al., 2020; Wang et al., 2019b; Honda et al., 2019). First applications of deep learning did not offer large improvements over more standard methods such as random forests (Wu et al., 2018; Jiang et al., 2021; Robinson et al., 2020). Consistent improvements were enabled by more efficient architectures adapted to this domain (Mayr et al., 2018; Yang et al., 2019a; Klicpera et al., 2020). In this spirit, our goal is to further advance modeling for any chemical task by re-designing self-attention for molecular data.

Encoding efficiently the relation between tokens in self-attention has been shown to substantially boost performance of Transformers in vision, language, music and biology (Shaw et al., 2018; Dai et al., 2019; Ingraham et al., 2021; Huang et al., 2020; Romero & Cordonnier, 2021; Khan et al., 2021; Ke et al., 2021). The vanilla self-attention includes absolute encoding of position, which can hinder learning when the absolute position in the sentence is not informative.[1] Relative positional encoding featurizes the relative distance between each pair of tokens, which led to substantial gains in the language and music domains (Shang et al., 2018; Huang et al., 2020). However, most Transformers for the chemical domain predominantly used no positional encoding in the self-

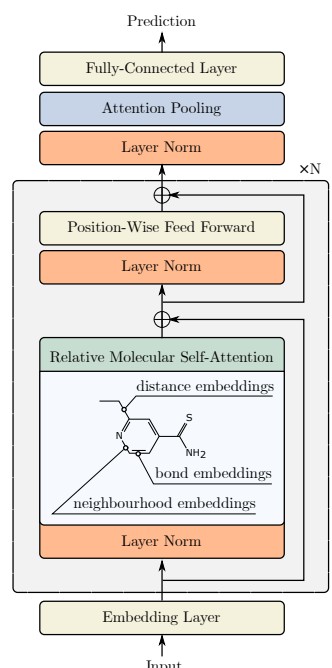

Prediction

Figure 1: Relative Molecule Attention Transformer uses a novel relative self-attention block tailored to molecule property prediction. It fuses three types of features: distance embedding, bond embedding and neighbourhood embedding.

attention layer (Chithrananda et al., 2020; Fabian et al., 2020; Rong et al., 2020; Wang et al., 2019b; Honda et al., 2019; Schwaller et al., 2019), which gives rise to similar issues with representing relations between atoms. We directly compare to (Maziarka et al., 2020), who introduced first self-attention module tailored to molecular data, and show large improvements across different tasks. Our work is also closely related to (Ingraham et al., 2021) that used relative self-attention fusing three dimensional structure with positional and graph based embedding, in the context of protein design.

---

[1]This arises for example when input is an arbitrary chunks of the text (Huang et al., 2020) (e.g. in the next sentence prediction task used in BERT pretraining).

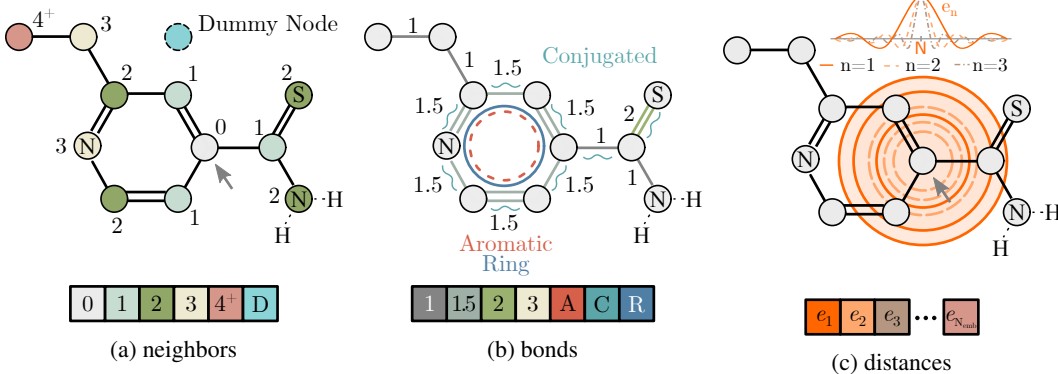

Figure 2: The Relative Molecule Self-Attention layer is based on the following features: (a) neighbourhood embedding one-hot encodes graph distances (neighbourhood order) from the source node marked with an arrow; (b) bond embedding one-hot encodes the bond order (numbers next to the graph edges) and other bond features for neighbouring nodes; (c) distance embedding uses radial basis functions to encode pairwise distances in the 3D space. These features are fused according to Equation (4).

## 3 RELATIVE MOLECULE SELF-ATTENTION

### 3.1 MOLECULAR SELF-ATTENTIONS

We first give a short background on how prior work on adapting self-attention for molecular data, and point out their potential shortcomings.

**Text Transformers** Multiple works have applied directly the Transformer to molecules encoded as text using the SMILES representation (Chithrananda et al., 2020; Fabian et al., 2020; Wang et al., 2019b; Honda et al., 2019; Schwaller et al., 2019). SMILES is a linear encoding of a molecule into a string of characters according to a deterministic ordering algorithm (Weininger, 1988; Jastrzębski et al., 2016). For example, the SMILES encoding of carbon dioxide is C(=O)=O.

Adding a single atom can completely change the ordering of atoms in the SMILES encoding. Hence, the relative positions of individual characters are not easily related to their proximity in the graph or space. This is in contrast to natural language processing, where the distance between two words in the sentence can be highly informative (Shaw et al., 2018; Huang et al., 2020; Ke et al., 2021). We suspect this makes the use of self-attention in SMILES models less effective. Another readily visible shortcoming is that the graph structure and distances between atoms of the molecule are either completely encoded or entirely thrown out.

**Graph Transformers** Several works have proposed Transformers that operate directly on a graph (Maziarka et al., 2020; Rong et al., 2020; Nguyen et al., 2019). The GROVER and the U2GNN models take as input a molecule encoded as a graph (Rong et al., 2020; Nguyen et al., 2019). In both of them, the self-attention layer does not have a direct access to the information about the graph. Instead, the information about the relations between atoms (existence of a bond or distance in the graph) is indirectly encoded by a graph convolutional layer that is run in GROVER within each layer, and in U2GNN only at the beginning. Similarly to Text Transformers, Graph Transformers also do not take into account the distances between atoms.

Structured Transformer introduced in (Ingraham et al., 2021) uses relative self-attention that operates on amino-acids in the task of protein design. Self-attention proposed by (Ingraham et al., 2021), similarly to our work, provides the model with information about the three dimensional structure of the molecule. As R-MAT encodes the relative distances between pairs of atoms, Structured Transformer also uses relative distances between modeled amino-acids. However, it encodes them in a slightly different way. We incorporate their ideas, and extend them to enable processing of molecular data.

**Molecule Attention Transformer** Our work is closely related to Molecule Attention Transformer (MAT), a transformer-based model with self-attention tailored to processing molecular data (Maziarka et al., 2020). In contrast to most of the aforementioned models, MAT incorporates the distance information in its self-attention module. MAT stacks $N$ Molecule Self-Attention blocks followed by a mean pooling and a prediction layer. For a $D$-dimensional state $\boldsymbol{x} \in \mathbb{R}^D$, the standard, vanilla self-attention operation is defined as

$$\mathcal{A}(\boldsymbol{x}) = \text{Softmax}\left(\frac{\mathbf{Q}\mathbf{K}^T}{\sqrt{d_k}}\right)\mathbf{V}, \tag{1}$$

where $\mathbf{Q} = \boldsymbol{x}\mathbf{W}^Q$, $\mathbf{K} = \boldsymbol{x}\mathbf{W}^K$, and $\mathbf{V} = \boldsymbol{x}\mathbf{W}^V$. Molecule Self-Attention extends Equation (1) to include additional information about bonds and distances between atoms in the molecule as

$$\mathcal{A}(\boldsymbol{x}) = \left(\lambda_a \, \text{Softmax}\left(\frac{\mathbf{Q}\mathbf{K}^T}{\sqrt{d_k}}\right) + \lambda_d \, g(\mathbf{D}) + \lambda_g \mathbf{A}\right)\mathbf{V}, \tag{2}$$

where $\lambda_a$, $\lambda_d$, $\lambda_g$ are the weights given to individual parts of the attention module, $g$ is a function given by either a softmax, or an element-wise $g(d) = \exp(-d)$, $\mathbf{A}$ is the adjacency matrix (with $\mathbf{A}_{(i,j)} = 1$ if there exists a bond between atoms $i$ and $j$ and 0 otherwise) and $\mathbf{D}$ is the distance matrix, where $\mathbf{D}_{(i,j)}$ represents the distance between the atoms $i$ and $j$ in the 3D space.

Self-attention can relate input elements in a highly flexible manner. In contrast, there is little flexibility in how Molecule Self-Attention can use the information about the distance between two atoms. The strength of the attention between two atoms depends monotonically on their relative distance. However, molecular properties can depend in a highly nonlinear way on the distance between atoms. This has motivated works such as (Klicpera et al., 2020) to explicitly model the interactions between atoms, using higher-order terms.

## 3.2 RELATIVE POSITIONAL ENCODING

In natural language processing, a vanilla self-attention layer does not take into account the positional information of the input tokens (i.e. if we permute the layer input, the output will stay the same). In order to add the positional information into the input data, a vanilla transformer enriches it with encoding of the absolute position. On the other hand, relative positional encoding (Shaw et al., 2018) adds the relative distance between each pair of tokens, which leads to substantial gains in the learned task. In our work, we use relative self-attention to encode the information about the relative neighbourhood, distances and physicochemical features between all pairs of atoms in the input molecule (See Figure 2).

## 3.3 ATOM RELATION EMBEDDING

Our core idea to improve Molecule Self-Attention is to add flexibility in how it processes graph and distance information. Specifically, we adapt positional relative encoding to processing molecules (Shaw et al., 2018; Dai et al., 2019; Huang et al., 2020; Ke et al., 2021), which we note was already hinted at in (Shaw et al., 2018) as a high-level future direction. The key idea in these works is to enrich the self-attention block to efficiently represent information about relative positions of items in the input sequence.

What reflects the relative position of two atoms in a molecule? Similarly to MAT, we delineate three inter-related factors: (1) their relative distance, (2) their distance in the molecular graph, and (3) their physiochemical relationship (e.g. are they within the same aromatic ring).

In the next step, we depart from Molecule Self-Attention (Maziarka et al., 2020) and introduce new factors to the relation embedding. Given two atoms, represented by vectors $\boldsymbol{x}_i, \boldsymbol{x}_j \in \mathbb{R}^D$, we encode their relation using an *atom relation embedding* $\boldsymbol{b}_{ij} \in \mathbb{R}^{D'}$. This embedding will then be used in the self-attention module after a projection layer. Next, we describe three components that are concatenated to form the embedding $\boldsymbol{b}_{ij}$.

**Neighbourhood embeddings** First, we encode the neighbourhood order between two atoms as a 6 dimensional one hot encoding, with information about how many other vertices are between nodes $i$ and $j$ in the original molecular graph (see Figure 2 and Table 4 from Appendix A).

**Distance embeddings** As we discussed earlier, we hypothesize that a much more flexible representation of the distance information should be facilitated in MAT. To achieve this, we use a radial basis distance encoding proposed by (Klicpera et al., 2020):

$$e_n(d) = \sqrt{\frac{2}{c}} \cdot \frac{\sin\left(\frac{n\pi}{c}d\right)}{d},$$

where $d$ is the distance between two atoms, $c$ is the predefined cutoff distance, $n \in \{1, \ldots, N_{emb}\}$ and $N_{emb}$ is the total number of radial basis functions that we use. Then obtained numbers are passed to the polynomial envelope function $u(d) = 1 - \frac{(p+1)(p+2)}{2}d^p + p(p+2)d^{p+1} - \frac{p(p+1)}{2}d^{p+2}$, with $p = 6$, in order to get the final distance embedding.

**Bond embeddings** Finally, we featurize each bond to reflect the physical relation between pairs of atoms that might arise from, for example, being part of the same aromatic structure in the molecule. Molecular bonds are embedded in as a 7 dimensional vector following (Coley et al., 2017) (see Table 5 from Appendix A). When the two atoms are not connected by a true molecular bond, all 7 dimensions are set to zeros. We note that while these features can be easily learned in pretraining, we hypothesize that this featurization might be highly useful for training R-MAT on smaller datasets.

## 3.4 RELATIVE MOLECULE SELF-ATTENTION

Equipped with the embedding $\boldsymbol{b}_{ij}$ for each pair of atoms in the molecule, we now use it to define a novel self-attention layer that we refer to as Relative Molecule Self-Attention.

First, mirroring the key-query-value design in the vanilla self-attention (c.f. Equation (1)), we transform $\boldsymbol{b}_{ij}$ into a key and value specific vectors $\boldsymbol{b}_{ij}^V, \boldsymbol{b}_{ij}^K$ using two neural networks $\phi_V$ and $\phi_K$. Each neural network consists of two layers. A hidden layer, shared between all attention heads and output layer, that create a separate relative embedding for different attention heads.

Consider Equation (1) in index notation:

$$\mathcal{A}(\boldsymbol{x})_i = \sum_{j=1}^{n} \text{Softmax}\left(\frac{e_{ij}}{\sqrt{d_z}}\right)^T (x_j W^V),$$

where the unnormalized attention is $e_{ij} = (x_i W^Q)(x_j W^K)^T$. By analogy, in Relative Molecule Self-Attention, we compute $e_{ij}$ as

$$e_{ij} = \underbrace{(x_i W^Q)(x_j W^K)^T}_{\text{vanilla self-attention}} + \underbrace{(x_i W^Q)\boldsymbol{b}_{ij}^K}_{\substack{\text{content-dependent} \\ \text{positional bias} \\ \text{for query}}} + \underbrace{(x_j W^K)\boldsymbol{b}_{ij}^K}_{\substack{\text{content-dependent} \\ \text{positional bias} \\ \text{for key}}} + \underbrace{\boldsymbol{u}^T(x_j W^K)}_{\substack{\text{global content} \\ \text{bias}}} + \underbrace{\boldsymbol{v}^T\boldsymbol{b}_{ij}^K}_{\substack{\text{global positional} \\ \text{bias}}}, \quad (3)$$

where $\boldsymbol{u}, \boldsymbol{v} \in \mathbb{R}^{D'}$ are trainable vectors. We then define Relative Molecule Self-Attention operation:

$$\mathcal{A}_i = \sum_{j=1}^{n} \text{Softmax}\left(\frac{e_{ij}}{\sqrt{d_z}}\right)^T (x_j W^V + \boldsymbol{b}_{ij}^V). \quad (4)$$

In other words, we enrich the self-attention layer with atom relations embedding. In the phase of attention weights calculation, we add content-dependent positional bias, global context bias and global positional bias (Dai et al., 2019; Huang et al., 2020) (that are calculated based on $\boldsymbol{b}_{ij}^K$) to the layer. Then, during calculation of the attention weighted average, we also include the information about the other embedding $\boldsymbol{b}_{ij}^V$.

### 3.5 RELATIVE MOLECULE ATTENTION TRANSFORMER

Finally, we use Relative Molecule Self-Attention to construct Relative Molecule Attention Transformer (R-MAT). The key changes compared to MAT are: (1) the use of Relative Molecule Self-Attention, (2) extended atom featurization, and (3) extended pretraining procedure. Figure 1 illustrates the R-MAT architecture.

The input is embedded as a matrix of size $N_{atom} \times 36$ where each atom of the input is embedded following (Coley et al., 2017; Pocha et al., 2020), see Table 6 of Appendix A. We process the input using $N$ stacked Relative Molecule Self-Attention attention layers. Each attention layer is followed by position-wise feed-forward Network (similar as in the classical transformer model (Vaswani et al., 2017)), which consists of 2 linear layers with a leaky-ReLU nonlinearity between them.

After processing the input using attention layers, we pool the representation into a constant-sized vector. We replace simple mean pooling with an attention-based pooling layer. After applying $N$ self-attention layers, we use the following self-attention pooling (Lin et al., 2017) in order to get the graph-level embedding of the molecule:

$$\begin{aligned} \mathbf{P} &= \text{Softmax}(W_2 \tanh(W_1 \mathbf{H}^T)), \\ \mathbf{g} &= \text{Flatten}(\mathbf{PH}), \end{aligned}$$

where $\mathbf{H}$ is the hidden state obtained from self-attention layers, $W_1 \in \mathbb{R}^{P \times D}$ and $W_2 \in \mathbb{R}^{S \times P}$ are pooling attention weights, with P equal to the pooling hidden dimension and S equal to the number of pooling attention heads. Finally, the graph embedding $\mathbf{g}$ is then passed to the two layer MLP, with leaky-ReLU activation in order to make the prediction.

**Pretraining** We used two-step pretraining procedure. In the first step, network is trained with the contextual property prediction task proposed by (Rong et al., 2020), where we mask not only selected atoms, but also their neighbours. The goal of the task is to predict the whole atom context. This task is much more demanding for the network than the classical masking approach presented by (Maziarka et al., 2020) since the network has to encode more specific information about the masked atom neighbourhood. Furthermore, the size of the context vocabulary is much bigger than the size of the atoms vocabulary in the MAT pretraining approach. The second task is a graph-level prediction proposed by (Fabian et al., 2020) in which the goal is to predict a set of real-valued descriptors of physicochemical properties. For more detailed information about the pretraining procedure and ablations, see Appendix B.

**Other details** Similarly to (Maziarka et al., 2020), we add an artificial dummy node to the input molecule. The distance of the dummy node to any other atom in the molecule is set to the maximal cutoff distance, and the edge connecting the dummy node with any other atom has its unique index (see index 5 in Table 4 of Appendix A). Moreover, the dummy node has its own index in the input atom embedding. We calculate distance information in the similar manner as (Maziarka et al., 2020). The 3D molecular conformations that are used to obtain distance matrices are calculated using UFFOPTIMIZEMOLECULE function from the RDKit package (Landrum, 2016) with the default parameters. Finally, we consider a variant of the model extended with 200 rdkit features as in (Rong et al., 2020). The features are concatenated to the final embedding $\mathbf{g}$ and processed using a prediction MLP.

## 4 EXPERIMENTS

### 4.1 SMALL HYPERPARAMETER BUDGET

The industrial drug discovery pipelines focus on fast iterations of compound screenings and adjusting the models to new data incoming from the laboratory. We start by comparing in this setting R-MAT to DMPNN (Yang et al., 2019a), MAT (Maziarka et al., 2020) and GROVER (Rong et al., 2020), representative state-of-the-art models on popular molecular property prediction tasks. We followed the evaluation in (Maziarka et al., 2020), where the only changeable hyperparameter is the learning rate, which was checked with 7 different values.

The BBBP and Estrogen-$\beta$ datasets use scaffold splits, while all the other datasets use random splits. Splits were proposed by (Maziarka et al., 2020). For every dataset we calculate scores based on 6 different splits, we report the mean test score based on the hyperparameters that obtained the best

validation score, in parentheses we include the standard deviation. In this and the next experiments, we denote models extended with additional rdkit features (see Section 3.5) as GROVER_rdkit and R-MAT _rdkit. More information about the models and datasets used in this benchmark are given in Appendix C.4.

Table 1 shows that R-MAT outperforms other methods in 3 out of 6 tasks. For comparison, we also cite representative results of other methods from (Maziarka et al., 2020). Satisfyingly, we observe a marked improvement on the solubility prediction tasks (ESOL and FreeSolv). Understanding solubility depends to a large degree on a detailed understanding of spatial relationships between atoms. This suggests that the improvement in performance might be related to better utilization of the distance or graph information.

Table 1: Results on molecule property prediction benchmark from (Maziarka et al., 2020). We only tune the learning rate for models in the first group. First two datasets are regression tasks (lower is better), other datasets are classification tasks (higher is better). For reference, we include results for non-pretrained baselines (SVM, RF, GCN (Duvenaud et al., 2015), and DMPNN (Yang et al., 2019a)) from (Maziarka et al., 2020). We also include SVM_rdkit and RF_rdkit as two baseline methods with added rdkit features. Rank-plot for these experiments is in Appendix D.1.

| | ESOL | FreeSolv | BBBP | Estrogen-$\beta$ | MetStab$_{low}$ | MetStab$_{high}$ |
|---|---|---|---|---|---|---|
| MAT | $.278_{(.020)}$ | $.265_{(.042)}$ | $.737_{(.009)}$ | $.773_{(.012)}$ | $.862_{(.025)}$ | $.884_{(.030)}$ |
| GROVER | $.303_{(.048)}$ | $.270_{(.033)}$ | $.726_{(.007)}$ | $.758_{(.006)}$ | $.892_{(.031)}$ | $.887_{(.019)}$ |
| GROVER$_{rdkit}$ | $.288_{(.021)}$ | $.308_{(.058)}$ | $.726_{(.003)}$ | $.788_{(.009)}$ | $.873_{(.033)}$ | $.881_{(.039)}$ |
| R-MAT | $.252_{(.030)}$ | $\mathbf{.232_{(.071)}}$ | $.745_{(.010)}$ | $.788_{(.007)}$ | $.887_{(.028)}$ | $.880_{(.027)}$ |
| R-MAT $_{rdkit}$ | $\mathbf{.246_{(.024)}}$ | $.239_{(.066)}$ | $\mathbf{.746_{(.007)}}$ | $.791_{(.010)}$ | $.884_{(.032)}$ | $.886_{(.031)}$ |
| SVM | $.479_{(.055)}$ | $.461_{(.077)}$ | $.723_{(.000)}$ | $.772_{(.000)}$ | $.893_{(.030)}$ | $\mathbf{.890_{(.029)}}$ |
| SVM$_{rdkit}$ | $.279_{(.024)}$ | $.285_{(.049)}$ | $.741_{(.001)}$ | $.781_{(.001)}$ | $.895_{(.029)}$ | $.884_{(.031)}$ |
| RF | $.534_{(.073)}$ | $.524_{(.098)}$ | $.721_{(.003)}$ | $.791_{(.012)}$ | $.892_{(.026)}$ | $.888_{(.030)}$ |
| RF$_{rdkit}$ | $.289_{(.035)}$ | $.337_{(.026)}$ | $.743_{(.002)}$ | $\mathbf{.807_{(.003)}}$ | $\mathbf{.903_{(.025)}}$ | $.886_{(.028)}$ |
| GCN | $.369_{(.032)}$ | $.299_{(.068)}$ | $.695_{(.013)}$ | $.730_{(.006)}$ | $.884_{(.033)}$ | $.875_{(.036)}$ |
| DMPNN | $.297_{(.046)}$ | $.252_{(.044)}$ | $.709_{(.001)}$ | $.776_{(.006)}$ | $.885_{(.026)}$ | $.889_{(.018)}$ |

### 4.2 LARGE HYPERPARAMETER BUDGET

In contrast to the previous setting, we test R-MAT against a similar set of models but using a large-scale hyperparameter search (300 different hyperparameter combinations). This setting has been proposed in (Rong et al., 2020). For comparison, we include results under small (7 different learning rates) hyperparameter budget. All datasets use a scaffold split. Scores are calculated based on 3 different data splits. While the ESOL and FreeSolv datasets are the same as in the previous paragraph, here they use a scaffold split and labels are not normalized (unlike in the previous paragraph). Additional information about the models and datasets used in this benchmark are given in Appendix C.5.

Table 2 summmarizes the experiment. Results show that for large hyperparameters budget R-MAT outperforms other methods in 2 tasks and along with GROVER are the best in one more task. In overall in this setting our method achieves comparable results to GROVER, having the same median rank and being slightly worse in terms of mean rank (see left side of Figure 7). On the other hand, for small hyperparameters budget R-MAT achieves the best results, both in terms of the mean and the median ranks (see right side of Figure 7).

### 4.3 LARGE-SCALE EXPERIMENTS

Finally, to better understand how R-MAT performs in a setting where pretraining is likely to less influence results, we include results on QM9 dataset (Ramakrishnan et al., 2014). QM9 is a quantum mechanics benchmark that encompasses prediction of 12 simulated properties across around 130k small molecules with at most 9 heavy (non-hydrogen) atoms. The molecules are provided with their atomic 3D positions for which the quantum properties were initially calculated. For these experiments, we used learning rate equal to 0.015 (we selected this learning rate value as it returned the best results for $\alpha$ dataset among 4 different learning rates that we tested: {0.005, 0.01, 0.015, 0.02}). Additional information about the dataset and models used in this benchmark are given in Appendix C.6

Table 2: Results on the benchmark from (Rong et al., 2020). Models are fine-tuned under a large hyperparameters budget. Additionally, models fine-tuned with only tuning the learning rate are presented in the last group. The last two datasets are classification tasks (higher is better), the remaining datasets are regression tasks (lower is better). For reference, we include results for non-pretrained baselines (GraphConv (Kipf & Welling, 2016), Weave (Kearnes et al., 2016) and DMPNN (Yang et al., 2019a)) from (Rong et al., 2020). We also include $RF_{rdkit}$ as a baseline method with added rdkit features. Rank-plot for these experiments is in Appendix D.2. We bold the best scores over all models and underline the best scores for learning rate tuned models only.

| | ESOL | FreeSolv | Lipo | QM7 | BACE | BBBP |
|---|---|---|---|---|---|---|
| $RF_{rdkit}$ | $.942_{(.196)}$ | $2.625_{(.509)}$ | $.739_{(.038)}$ | $124.3_{(3.5)}$ | $\mathbf{.884}_{(\mathbf{.030})}$ | $.928_{(.025)}$ |
| GraphConv | $1.068_{(.050)}$ | $2.900_{(.135)}$ | $.712_{(.049)}$ | $118.9_{(20.2)}$ | $.854_{(.011)}$ | $.877_{(.036)}$ |
| Weave | $1.158_{(.055)}$ | $2.398_{(.250)}$ | $.813_{(.042)}$ | $94.7_{(2.7)}$ | $.791_{(.008)}$ | $.837_{(.065)}$ |
| DMPNN | $.980_{(.258)}$ | $2.177_{(.914)}$ | $.653_{(.046)}$ | $105.8_{(13.2)}$ | $.852_{(.053)}$ | $.919_{(.030)}$ |
| GROVER$_{rdkit}$ | $.888_{(.116)}$ | $\mathbf{1.592}_{(\mathbf{.072})}$ | $\mathbf{.563}_{(\mathbf{.030})}$ | $72.5_{(5.9)}$ | $.878_{(.016)}$ | $\mathbf{.936}_{(\mathbf{.008})}$ |
| R-MAT $_{rdkit}$ | $\mathbf{.786}_{(\mathbf{.133})}$ | $2.044_{(.662)}$ | $.574_{(.028)}$ | $\mathbf{68.692}_{(\mathbf{1.123})}$ | $.871_{(.028)}$ | $\mathbf{.936}_{(\mathbf{.020})}$ |
| MAT | $.853_{(.159)}$ | $\underline{1.744}_{(.425)}$ | $.608_{(.017)}$ | $102.8_{(2.94)}$ | $.846_{(.025)}$ | $.920_{(.039)}$ |
| GROVER | $.927_{(.110)}$ | $2.262_{(.407)}$ | $.604_{(.015)}$ | $82.623_{(3.833)}$ | $.867_{(.022)}$ | $.908_{(.053)}$ |
| GROVER$_{rdkit}$ | $.924_{(.129)}$ | $2.096_{(.496)}$ | $.593_{(.029)}$ | $84.625_{(4.174)}$ | $\underline{.873}_{(.031)}$ | $.931_{(.021)}$ |
| R-MAT | $\underline{.801}_{(.132)}$ | $1.912_{(0.364)}$ | $.585_{(.029)}$ | $77.248_{(2.819)}$ | $.858_{(.041)}$ | $\underline{.931}_{(.016)}$ |
| R-MAT $_{rdkit}$ | $.819_{(.145)}$ | $2.057_{(.434)}$ | $\underline{.580}_{(.019)}$ | $70.929_{(3.568)}$ | $.858_{(.021)}$ | $.920_{(.021)}$ |

Figure 3 compares R-MAT performance with various models. More detailed results could be find in Table 10 from Appendix D.3. R-MAT achieves highly competitive results, with state-of-the-art performance on 4 out of the 12 tasks. We attribute higher variability of performance to the limited small hyperparameter search we performed.

### 4.4 EXPLORING THE DESIGN SPACE OF SELF-ATTENTION LAYER

Achieving strong empirical results hinged on a methodologically exploration the design space of different variants of the self-attention layer. We document here this exploration and relevant ablations. Due to space limitations, we defer most results to the Appendix E. We perform all experiments on the ESOL, FreeSolv and BBBP datasets with 3 different scaffold splits. We did not use any pretraining for these experiments. We follow the same fine-tuning methodology as in Section 4.1.

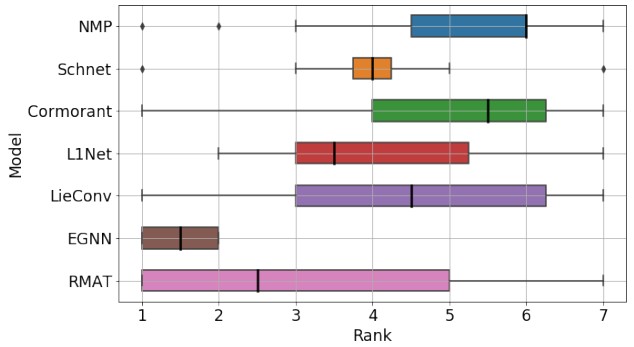

Figure 3: Rank plot of scores obtained on the QM9 benchmark, which consists of 12 different quantum property prediction tasks.

**Importance of different sources of information in self-attention** The self-attention module in R-MAT incorporates three auxiliary sources of information: (1) distance information, (2) graph information (encoded using neighbourhood order), and (3) bond features. In Table 3 (Left), we show the effect on performance of ablating each of this elements. Importantly, we find that each component is important to R-MAT performance, including the distance matrix.

**Maximum neighbourhood order** We take a closer look at how we encode the molecular graph. (Maziarka et al., 2020) used a simple binary adjacency matrix to encode the edges. We enriched this representation by adding one-hot encoding of the neighbourhood order. For example, the order of 3 for a pair of atoms means that there are two other vertices on the shortest path between this pair of atoms. In R-MAT we used 4 as the maximum order of neighbourhood distance. That is, we encoded as separate features if two atoms are 1, 2, 3 or 4 *hops* away in the molecular graph. In Table 3 (Right)

Table 3: Ablations of Relative Molecule Self-Attention; other ablations are included in the Appendix.

(a) Test set performances of R-MAT for different relative attention features.

|  | BBBP | ESOL | FreeSolv |
|---|---|---|---|
| R-MAT | $.908_{(.039)}$ | $.378_{(.027)}$ | $.438_{(.036)}$ |
| distance | $.858_{(.064)}$ | $.412_{(.038)}$ | $.468_{(.022)}$ |
| neighbourhood | $.867_{(.043)}$ | $.390_{(.020)}$ | $.545_{(.023)}$ |
| bond features | $.860_{(.032)}$ | $.395_{(.020)}$ | $.536_{(.035)}$ |

(b) Test set performances of R-MAT for different choices of maximum neighbourhood order.

|  | BBBP | ESOL | FreeSolv |
|---|---|---|---|
| R-MAT | $.908_{(.039)}$ | $.378_{(.027)}$ | $.438_{(.036)}$ |
| Max order = 1 | $.847_{(.081)}$ | $.372_{(.018)}$ | $.461_{(.049)}$ |
| Max order = 2 | $.890_{(.068)}$ | $.382_{(.040)}$ | $.519_{(.036)}$ |
| Max order = 3 | $.873_{(.053)}$ | $.455_{(.005)}$ | $.492_{(.055)}$ |

we ablate this choice. The result suggests that R-MAT performance benefits from including separate feature for all the considered orders.

**Closer comparison to Molecule Attention Transformer** Our main motivation for improving self-attention in MAT was to make it easier to represent attention patterns that depend in a more complex way on the distance and graph information. We qualitatively explore here whether R-MAT achieves this goal, comparing its attention patterns to that of MAT. From the Figure 4 one can see that indeed R-MAT seems capable of learning more complex attention patterns than MAT. We add a more detailed comparison, with more visualised attention heads in Appendix D.4.

## 5 CONCLUSIONS

Transformer has been successfuly adapted to various domain by incorporating into its architecture a minimal set of inductive biases. In a similar spirit, we methodologically explored the design space of the self-attention layer, and identified a highly effective Relative Molecule Self-Attention.

Relative Molecule Attention Transformer, a model based on Relative Molecule Self-Attention, achieves state-of-the-art or very competitive results across a wide range of

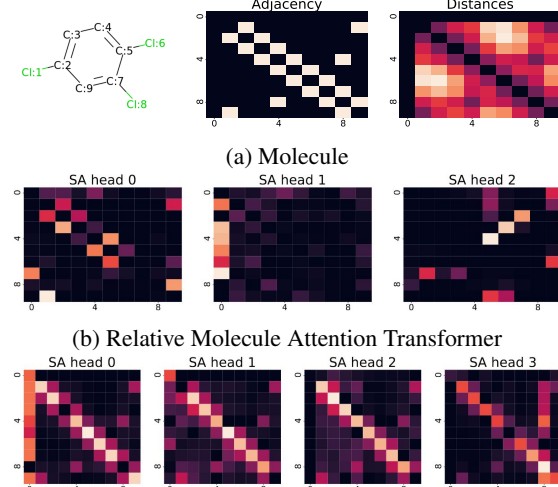

(a) Molecule

(b) Relative Molecule Attention Transformer

(c) Molecule Attention Transformer

Figure 4: Visualization of the learned self-attention for each of the first 3 attention heads in the second layer of pretrained R-MAT (middle) and the first 4 attention heads in pretrained MAT (bottom), for a molecule from the ESOL dataset. The top Figure visualizes the molecule and its adjacency and distance matrices. The self-attention pattern in MAT is dominated by the adjacency and distance matrix, while R-MAT seems capable of learning more complex attention patterns.

molecular property prediction tasks. R-MAT is a highly versatile model, showing state-of-the-art results in both quantum property prediction tasks, as well as on biological datasets. We also show that R-MAT is easy to train and requires tuning only the learning rate to achieve competitive results, which together with open-sourced weight and code, makes it is highly accessible.

Relative Molecule Self-Attention encodes an inductive bias to consider relationships between atoms that are commonly relevant to a chemist, but on the other hand leaves flexibility to unlearn them if needed. Relatedly, Vision Transformers learn global processing in early layers despite being equipped with a locality inductive bias (Dosovitskiy et al., 2021). Our empirical results show in a new context that picking the right set of inductive biases is key for self-supervised learning to work well. We also how that Relative Molecule Self-Attention will help improve other models for molecular property prediction.

Learning useful representations for molecular property prediction is far from solved. Achieving state-of-the-art results, while less dependent on them, still relied on using certain large sets of handcrafted features both in fine-tuning and pretraining. At the same time, these features are beyond doubt learnable from data. Developing methods that will push representation learning towards discovering these and better features automatically from data is an exciting challenge for the future.

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

# A R-MAT NODE AND EDGE FEATURES

In the following section, we present the node and edge features used by R-MAT.

## A.1 EDGE FEATURES

In R-MAT, all atoms are connected with an edge. The vector representation of every edge contains information about atoms neighbourhood, distances between them and physicochemical features of a bond if it exists (see Figure 2)

**Neighbourhood embeddings**   The neighbourhood information of an atom pair is represented by a 6-dimensional one-hot encoded vector, with features presented in Table 4. Every neighbourhood embedding contains the information about how many other vertices are between nodes $i$ and $j$ in the original molecular graph.

Table 4: Featurization used to embed neighbourhood order in R-MAT.

| Indices | Description |
|---------|-------------|
| 0 | $i = j$ |
| 1 | Atoms $i$ and $j$ are connected with a bond |
| 2 | In the shortest path between atoms $i$ and $j$ there is one atom |
| 3 | In the shortest path between atoms $i$ and $j$ there are two atoms |
| 4 | In the shortest path between atoms $i$ and $j$ there are three or more atoms |
| 5 | Any of the atoms $i$ or $j$ is a dummy node |

**Bond embeddings**   Molecular bonds are embedded in a 7-dimensional vector following (Coley et al., 2017), with features specified in Table 5. When the two atoms are not connected by a true molecular bond, all 7 dimensions are set to zeros.

Table 5: Featurization used to embed molecular bonds in R-MAT.

| Indices | Description |
|---------|-------------|
| $0 - 3$ | Bond order as one-hot vector of 1, 1.5, 2, 3 |
| 4 | Is aromatic |
| 5 | Is conjugated |
| 6 | Is in a ring |

## A.2 NODE FEATURES

The input molecule is embedded as a matrix of size $N_{atom} \times 36$ where each atom of the input is embedded following (Coley et al., 2017; Pocha et al., 2020). All features are presented in Table 6.

# B PRETRAINING

We extend the pretraining procedure of (Maziarka et al., 2020), who used a masking task based on (Devlin et al., 2018; Hu et al., 2020); they masked types of some of the graph atoms and treat them as the label, that should be predicted by the neural network. Such approach works well in NLP where models pretrained with the masking task create the state-of-the-art representation (Devlin et al., 2018; Liu et al., 2019; Yang et al., 2019b). However in chemistry, otherwise than in NLP, the size of atoms vocabulary is much smaller. Moreover, usually only one type of atom fits a given place and thus the

Table 6: Featurization used to embed atoms in R-MAT.

| Indices | Description |
|---|---|
| $0 - 11$ | Atomic identity as a one-hot vector of B, N, C, O, F, P, S, Cl, Br, I, Dummy, other |
| $12 - 17$ | Number of heavy neighbors as one-hot vector of 0, 1, 2, 3, 4, 5 |
| $18 - 22$ | Number of hydrogen atoms as one-hot vector of 0, 1, 2, 3, 4 |
| $23 - 33$ | Formal charge as one-hot vector of -5, -4, ..., 4, 5 |
| 34 | Is in a ring |
| 35 | Is aromatic |

representation trained with the masking task has problems with encoding meaningful information in chemistry.

## B.1 CONTEXTUAL PRETRAINING

Instead of atom masking, we used a two-step pretraining that combines the procedures proposed by (Rong et al., 2020; Fabian et al., 2020). In the first step, the network is trained with the contextual property prediction task (Rong et al., 2020), where we mask not only the selected atoms, but also their neighbours. The task is then to predict the whole atom context, e.g. if the selected atom's type is carbon connected with a nitrogen with a double bond and with an oxygen with a single bond, we encode the atom neighbourhood as C_N-DOUBLE1_O-SINGLE1 (we list all the node-edge counts terms in the alphabetical order), then the network has to predict the specific type of the masked neighbourhood for every masked atom. This task is much more demanding for the network than the classical masking approach presented by (Maziarka et al., 2020) as the network has to encode more specific information about the masked atom's neighbourhood. Furthermore, the size of context vocabulary is much bigger than the size of atoms vocabulary in the MAT pretraining approach (2925 for R-MAT vs 35 for MAT).

## B.2 GRAPH-LEVEL PRETRAINING

The second task is the graph-level property prediction proposed by (Fabian et al., 2020). In this pretraining procedure, the task is to predict 200 real-valued descriptors of physicochemical characteristics of every given molecule.

The list of all 200 descriptors from RDKit is as follows:
```
BalabanJ, BertzCT, Chi0, Chi0n, Chi0v, Chi1, Chi1n, Chi1v, Chi2n,
Chi2v, Chi3n, Chi3v, Chi4n, Chi4v, EState_VSA1, EState_VSA10,
EState_VSA11, EState_VSA2, EState_VSA3, EState_VSA4, EState_VSA5,
EState_VSA6, EState_VSA7, EState_VSA8, EState_VSA9, ExactMolWt,
FpDensityMorgan1, FpDensityMorgan2, FpDensityMorgan3, FractionCSP3,
HallKierAlpha, HeavyAtomCount, HeavyAtomMolWt, Ipc, Kappa1, Kappa2,
Kappa3, LabuteASA, MaxAbsEStateIndex, MaxAbsPartialCharge, MaxEStateIndex,
MaxPartialCharge, MinAbsEStateIndex, MinAbsPartialCharge, MinEStateIndex,
MinPartialCharge, MolLogP, MolMR, MolWt, NHOHCount, NOCount,
NumAliphaticCarbocycles, NumAliphaticHeterocycles, NumAliphaticRings,
NumAromaticCarbocycles, NumAromaticHeterocycles, NumAromaticRings,
NumHAcceptors, NumHDonors, NumHeteroatoms, NumRadicalElectrons,
NumRotatableBonds, NumSaturatedCarbocycles, NumSaturatedHeterocycles,
NumSaturatedRings, NumValenceElectrons, PEOE_VSA1, PEOE_VSA10,
PEOE_VSA11, PEOE_VSA12, PEOE_VSA13, PEOE_VSA14, PEOE_VSA2, PEOE_VSA3,
PEOE_VSA4, PEOE_VSA5, PEOE_VSA6, PEOE_VSA7, PEOE_VSA8, PEOE_VSA9,
RingCount, SMR_VSA1, SMR_VSA10, SMR_VSA2, SMR_VSA3, SMR_VSA4, SMR_VSA5,
```

```
SMR_VSA6, SMR_VSA7, SMR_VSA8, SMR_VSA9, SlogP_VSA1, SlogP_VSA10,
SlogP_VSA11, SlogP_VSA12, SlogP_VSA2, SlogP_VSA3, SlogP_VSA4, SlogP_VSA5,
SlogP_VSA6, SlogP_VSA7, SlogP_VSA8, SlogP_VSA9, TPSA, VSA_EState1,
VSA_EState10, VSA_EState2, VSA_EState3, VSA_EState4, VSA_EState5,
VSA_EState6, VSA_EState7, VSA_EState8, VSA_EState9, fr_Al_COO, fr_Al_OH,
fr_Al_OH_noTert, fr_ArN, fr_Ar_COO, fr_Ar_N, fr_Ar_NH, fr_Ar_OH,
fr_COO, fr_COO2, fr_C_O, fr_C_O_noCOO, fr_C_S, fr_HOCCN, fr_Imine,
fr_NH0, fr_NH1, fr_NH2, fr_N_O, fr_Ndealkylation1, fr_Ndealkylation2,
fr_Nhpyrrole, fr_SH, fr_aldehyde, fr_alkyl_carbamate, fr_alkyl_halide,
fr_allylic_oxid, fr_amide, fr_amidine, fr_aniline, fr_aryl_methyl,
fr_azide, fr_azo, fr_barbitur, fr_benzene, fr_benzodiazepine,
fr_bicyclic, fr_diazo, fr_dihydropyridine, fr_epoxide, fr_ester, fr_ether,
fr_furan, fr_guanido, fr_halogen, fr_hdrzine, fr_hdrzone, fr_imidazole,
fr_imide, fr_isocyan, fr_isothiocyan, fr_ketone, fr_ketone_Topliss,
fr_lactam, fr_lactone, fr_methoxy, fr_morpholine, fr_nitrile, fr_nitro,
fr_nitro_arom, fr_nitro_arom_nonortho, fr_nitroso, fr_oxazole,
fr_oxime, fr_para_hydroxylation, fr_phenol, fr_phenol_noOrthoHbond,
fr_phos_acid, fr_phos_ester, fr_piperdine, fr_piperzine, fr_priamide,
fr_prisulfonamd, fr_pyridine, fr_quatN, fr_sulfide, fr_sulfonamd,
fr_sulfone, fr_term_acetylene, fr_tetrazole, fr_thiazole, fr_thiocyan,
fr_thiophene, fr_unbrch_alkane, fr_urea, qed
```

## C  EXPERIMENTAL SETTING

### C.1  MODEL HYPERPARAMETERS

R-MAT model consists of 10 layers with 12 attention heads in each, $d_{model} = 768$. The distance layer consists of 32 radial functions ($N_{emb} = 32$) and cutoff distance $c$ is set to 20 Å. Attention pooling consists of 4 pooling heads and pooling hidden dimension is set to 128. Prediction MLP consists of one hidden layer with dimension set to 1024 and dropout 0.1. R-MAT uses leaky-ReLU with slope 0.1 as a non-linearity in all experiments. This set of hyperparameters defines a model with 48 millions of parameters, which is equal to the number of parameters of GROVER$_{base}$ and is slightly higher than the number of parameters of MAT (42M).

### C.2  3D CONFORMATIONS

The 3D molecular conformations that are used to obtain distance matrices were calculated using UFFOPTIMIZEMOLECULE function from the RDKit package (Landrum, 2016) with the default parameters (MAXITERS=200, VDWTHRESH=10.0, CONFID=−1, IGNOREINTERFRAGINTERACTIONS=True).

One disadvantage of this approach is the costly calculation of the distance matrix. Which is not burdensome for small datasets with small molecules (e.g FreeSolv), however, they can be a problem with larger ones (e.g. dataset for pretraining). In Table 7 we present dataset sizes and molecular statistics for three different datasets (FreeSolv, ESOL, BBBP), as well as time needed for molecular conformations calculation. In Figure 5 we present these times for every single molecule from these datasets. Based on these results one can see, that the larger the molecule, the longer the conformation calculation time. Moreover even for BBBP, that is not rather a large dataset, calculating the conformations could take almost 5 minutes. We leave exploring other conformation methods (more accurate or faster) as an interesting topic for the future.

Table 7: Time needed for molecular conformations calculation for different datasets.

|          | Dataset size | Average number of atoms | Calculation time (s) | Average calculation time (s) |
|----------|--------------|-------------------------|----------------------|------------------------------|
| FreeSolv | 622          | 8.72                    | 5.225                | 0.008                        |
| ESOL     | 1128         | 13.28                   | 27.865               | 0.024                        |
| BBBP     | 2037         | 24.02                   | 267.084              | 0.131                        |

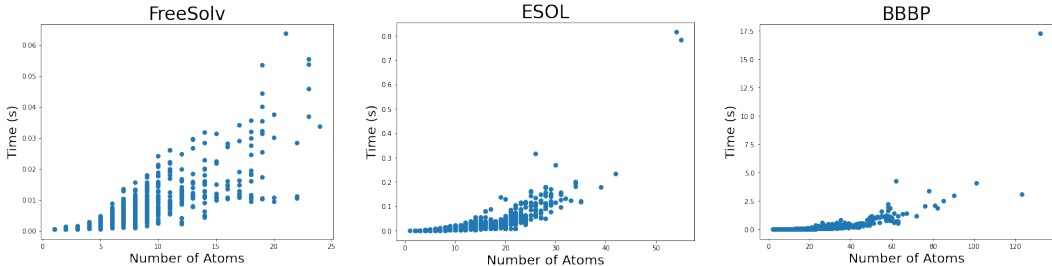

Figure 5: Time needed for conformations calculation for every single molecule for different datasets.

## C.3 PRETRAINING

We pretrained R-MAT on 4 millions of unlabelled molecules. Molecules were obtained from the ZINC15 (Sterling & Irwin, 2015) and ChEMBL (Gaulton et al., 2011) datasets, by first taking a sample of 10M molecules and then filtering them using the Lipinski's rule of five (Lipinski et al., 1997). We have split the data into training and validation datasets, where validation dataset consists of 5% of the data. We pretrained R-MAT for 150 epochs. We used dropout equal to 0.1, learning rate 0.001, the Noam optimizer (Vaswani et al., 2017) trained with 20000 warm-up steps and batch size 256. We pretrained R-MAT with 8 Nvidia A100 GPUs, using the Horovod package (Sergeev & Del Balso, 2018).

**Overlap between pre-trained and task datasets** In Table 8 we provide the information about number of atoms that overlaps between pre-trained dataset and all tasks datasets used in this paper. One can see that usually the overlap is at most few percent of all dataset molecules. This overlap does not affect the test performance, as pre-training consists of semi-supervised task of context prediction and other task, associated with the prediction of simple physico-chemical characteristics, loosely related to the fine-tuning tasks, which are additionally added to the final graph embedding during the prediction.

Table 8: Overlap between pre-trained dataset and different tasks datasets.

| dataset | overlap molecules | overlap percentage |
|---|---|---|
| bbbp | 34 | 1.6% |
| bace | 0 | 0.0% |
| esol | 83 | 7.3% |
| estrogen-beta | 45 | 2.2% |
| freesolv | 2 | 0.3% |
| lipo | 128 | 3.0% |
| mesta-high | 608 | 28.5% |
| mesta-low | 608 | 28.5% |
| qm7 | 0 | 0.0% |
| qm9 | 0 | 0.0% |

## C.4 SMALL HYPERPARAMETER BUDGET

**Models** We compared R-MAT with four models trained from scratch: Support Vector Machine with RBF kernel (SVM) and Random Forest (RF) that both works on ECFP fingerprints (Rogers & Hahn, 2010), Graph Convolutional Network (Duvenaud et al., 2015) (GCN) and Directed Message Passing Neural Network (Yang et al., 2019a) (DMPNN).

The comparison also includes two different pretrained models: MAT (Maziarka et al., 2020) and GROVER (Rong et al., 2020).

**Datasets** The benchmark is based on important molecule property tasks in the drug discovery domain. The first two datasets are ESOL and FreeSolv, in which the task is to predict the solubility of

a molecule in water – a key property of any drug – and the error is measured using RMSE. The goal in BBBP and Estrogen$-\beta$ is to classify correctly whether a given molecule is active against a biological target. For details on other tasks please see (Maziarka et al., 2020). BBBP and Estrogen$-\beta$ used scaffold split, rest datasets used random split method. For every dataset 6 different splits were created. Labels of regression datasets (ESOL and FreeSolv) were normalized before training. We did not include the Estrogen$-\alpha$ dataset that was also presented by (Maziarka et al., 2020), due to the GPU memory limitations (as the biggest molecule from this dataset consists of over 500 atoms).

**Training hyperparameters**  We fine-tune R-MAT on the target tasks for 100 epochs, with batch size equal to 32 and Noam optimizer with warm-up equal to 30% of all steps. The only hyperparameter that we tune is the learning rate, which is selected from the set of 7 possible options: $\{1e-3, 5e-4, 1e-4, 5e-5, 1e-5, 5e-6, 1e-6\}$. This small budget for hyperparameter selection reflects the long-term goal of this paper of developing easy to use models for molecule property prediction. The fine-tuning was conducted using Nvidia V100 GPU.

### C.5 LARGE HYPERPARAMETER BUDGET

**Models**  For large hyperparameters budget we compared R-MAT with three models trained from scratch: GraphConv (Kipf & Welling, 2016), Weave (Kearnes et al., 2016) and DMPNN (Yang et al., 2019a) and two pre-trained models: MAT (Maziarka et al., 2020) and GROVER (Rong et al., 2020). For small hyperparameters budget we compared R-MAT to MAT and GROVER. We note that R-MAT, MAT and GROVER use different pretraining methods. MAT was pretrained with 2M molecules from ZINC database and GROVER was pretrained with 10M molecules from the ZINC and ChEMBL databases.

**Datasets**  All datasets were splitted using a scaffold split. The resulting splits are different than in the MAT benchmark. For every dataset, 3 different splits were created. In our comparison we included only the subset of single-task datasets from original GROVER work (Rong et al., 2020). This is the reason why we use a smaller number of datasets. The obtained regression scores for ESOL differs significantly from the small hyperparameters budget benchmark because this time labels are not normalized.

**Training hyperparameters**  For learning rate tuning we used the same hyperparameters settings as in the MAT benchmark (see Appendix C.4).

For large hyperparameters budget we run random search with the hyperparameters listed in a Table 9.

Table 9: Relative Molecule Attention Transformer large grid hyperparameters ranges

|  | parameters |
| --- | --- |
| warmup | 0.05, 0.1, 0.2, 0.3 |
| learning rate | 0.005, 0.001, 0.0005, 0.0001, 0.00005, 0.00001, 0.000005, 0.000001 |
| epochs | 100 |
| pooling hidden dimension | 64, 128, 256, 512, 1024 |
| pooling attention heads | 2, 4, 8 |
| prediction MLP layers | 1, 2, 3 |
| prediction MLP dim | 256, 512, 1024, 2048 |
| prediction MLP dropout | 0.0, 0.1, 0.2 |

### C.6 LARGE-SCALE EXPERIMENTS

**Models**  We compared our R-MAT with 8 different models: NMP (Gilmer et al., 2017), Schnet (Schütt et al., 2017), Cormorant (Anderson et al., 2019), L1Net (Miller et al., 2020), LieConv (Finzi et al., 2020), TFN (Thomas et al., 2018), SE(3)-Tr. (Fuchs et al., 2020), EGNN (Satorras et al., 2021).

**Datasets** The QM9 dataset (Ramakrishnan et al., 2014) is a dataset that consists of molecules, with up to 9 heavy atoms (H, C, N, O, F) and up to 29 atoms overall per molecule. Each atom from this dataset is additionally associated with 3D position. The dataset consists of 12 different regression tasks named: $\alpha, \Delta\varepsilon, \varepsilon_{\text{HOMO}}, \varepsilon_{\text{LUMO}}, \mu, C_\nu, G, H, R^2, U, U_0$, ZPVE, for which the mean absolute error is a standard metric. The dataset has over 130k molecules. We use data splits proposed by (Anderson et al., 2019), which gives us 100k trainin molecules, 18k molecules for validation and 13k molecules for testing.

**Training hyperparameters** We trained R-MAT for 1000 epochs, with batch size equal to 256 and learning rate equal to 0.015. We report the test set MAE for the epoch with the lowest validation MAE. We selected this learning rate value as it returned the best results for $\alpha$ among 4 different learning rates that we tested: $\{0.005, 0.01, 0.015, 0.02\}$.

### C.7 ABLATIONS

**Datasets** For the ablations section, we used the BBBP, ESOL and FreeSolv datasets, splitted using a scaffold split, with 3 different splits. Labels of the regression datasets (ESOL and FreeSolv) were normalized before training. Scores obtained in this section differ significantly from the previous benchmarks due to the different data splits, different model hyperparameters and no pretraining used.

**Training hyperparameters** Similarly as for our main benchmarks, we tuned only the learning rate, which was selected from the set of 7 possible options: $\{1e{-}3, 5e{-}4, 1e{-}4, 5e{-}5, 1e{-}5, 5e{-}6, 1e{-}6\}$. We used batch size equal to 32 and Noam optimizer with warm-up equal to 20% of all steps. Moreover we use single layer, instead of two-layers MLP as our classification part.

## D ADDITIONAL EXPERIMENTAL RESULTS

### D.1 SMALL HYPERPARAMETER BUDGET

In Figure 6 one can find rank plots for results from Table 1. R-MAT and R-MAT $_{\text{rdkit}}$ obtained the best median rank among all compared models. The performance gain of RF$_{\text{rdkit}}$ over RF and SVM$_{\text{rdkit}}$ over SVM is interesting. Overall these two baseline models are worse only than R-MAT.

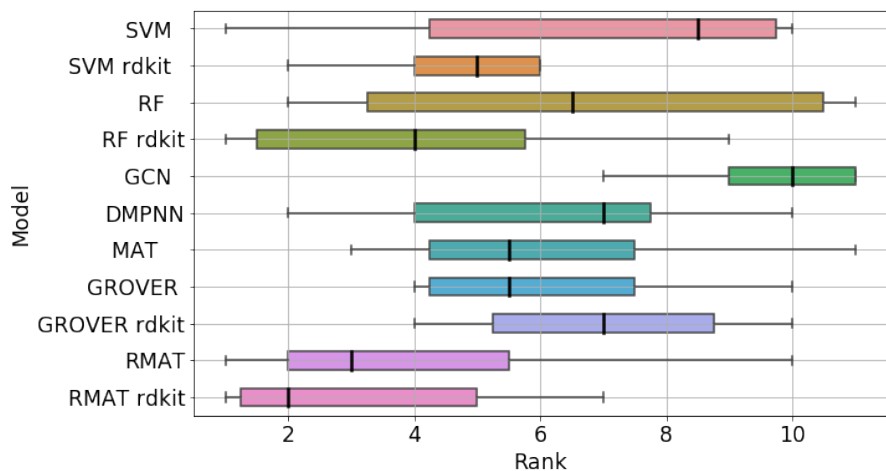

Figure 6: Rank plot for small hyperparameter budget experiments.

### D.2 LARGE HYPERPARAMETER BUDGET

In Figure 7 one can find rank plots for results from Table 2. We present separate plots for models trained with the large grid search (Left) and for models with only learning rate tuning (Right).

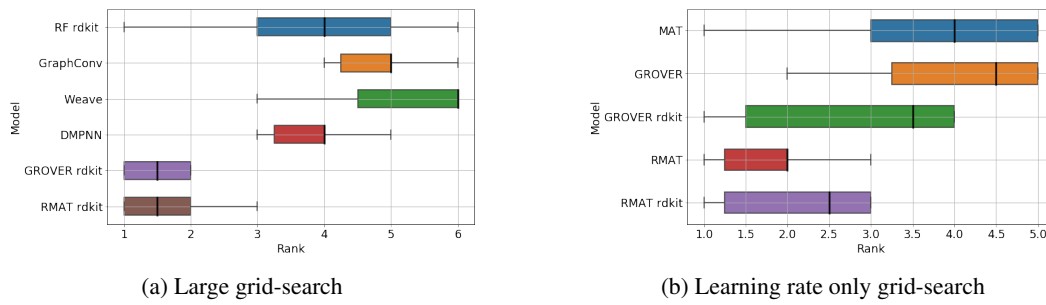

(a) Large grid-search            (b) Learning rate only grid-search

Figure 7: Rank plot for large hyperparameter budget (Left) as well as for models trained with only the learning rate tunning (Right).

### D.3 LARGE-SCALE EXPERIMENTS

In Table 10 one can find detailed results of comparison R-MAT performance with other various models. R-MAT achieves highly competitive results, with state-of-the-art performance on 4 out of the 12 tasks, which proves how universal this model is.

Table 10: Mean absolute error on QM9, a benchmark including various quantum prediction tasks. Results are cited from literature.

| Task | $\alpha$ | $\Delta\varepsilon$ | $\varepsilon_{\text{HOMO}}$ | $\varepsilon_{\text{LUMO}}$ | $\mu$ | $C_\nu$ | $G$ | $H$ | $R^2$ | $U$ | $U_0$ | ZPVE |
| Units | bohr$^3$ | meV | meV | meV | D | cal/mol K | meV | meV | bohr$^3$ | meV | meV | meV |
|---|---|---|---|---|---|---|---|---|---|---|---|---|
| NMP | .092 | 69 | 43 | 38 | .030 | .040 | 19 | 17 | .180 | 20 | 20 | 1.50 |
| Schnet | .235 | 63 | 41 | 34 | .033 | .033 | 14 | 14 | .073 | 19 | 14 | 1.70 |
| Cormorant | .085 | 61 | 34 | 38 | .038 | .026 | 20 | 21 | .961 | 21 | 22 | 2.03 |
| L1Net | .088 | 68 | 46 | 35 | .043 | .031 | 14 | 14 | .354 | 14 | 13 | 1.56 |
| LieConv | .084 | 49 | 30 | 25 | .032 | .038 | 22 | 24 | .800 | 19 | 19 | 2.28 |
| TFN | .223 | 58 | 40 | 38 | .064 | .101 | - | - | - | - | - | - |
| SE(3)-Tr. | .142 | 53 | 35 | 33 | .051 | .054 | - | - | - | - | - | - |
| EGNN | .071 | 48 | 29 | 25 | .029 | .031 | 12 | 12 | .106 | 12 | 11 | 1.55 |
| R-MAT $_{\text{rdkit}}$ | .082 | 48 | 31 | 29 | .110 | .036 | 10 | 10 | .676 | 10 | 12 | 2.23 |

### D.4 CLOSER COMPARISON TO MOLECULE ATTENTION TRANSFORMER

Our main motivation for improving self-attention in MAT was to make it easier to represent attention patterns that depend in a more complex way on the distance and graph information. We qualitatively explore here whether R-MAT achieves this goal, comparing its attention patterns to that of MAT.

For this purpose we compared attention patterns learned by pretrained MAT (weights taken from (Maziarka et al., 2020)) and R-MAT for a selected molecule from the ESOL dataset. Figure 8 shows that different heads of Relative Molecule Self-Attention are focusing on different atoms in the input molecule. We can see that self-attention strength is concentrated on the input atom (head 5), on the closest neighbours (heads 0 and 11), on the second order neighbours (head 7), on the dummy node (head 1) or on some substructure that occurs in the molecule (heads 6 and 10 are concentrated on atoms 1 and 2). In contrast, self-attention in MAT focuses mainly on the input atoms and its closest neighbours, the information from other regions of the molecule is not strongly propagated. This likely happens due to the construction of the Molecule Self-Attention in MAT (c.f. Equation (2)), where the output atom representation is calculated from equally weighted messages based on the adjacency matrix, distance matrix and self-attention. Due to its construction, it is more challenging for MAT than for R-MAT to learn to attend to a distant neighbour.

### E EXPLORING THE DESIGN SPACE OF MOLECULAR SELF-ATTENTION

Identifying the Relative Molecule Self-Attention layer required a large-scale and methodological exploration of the self-attention design space. In this section, we present experimental data that

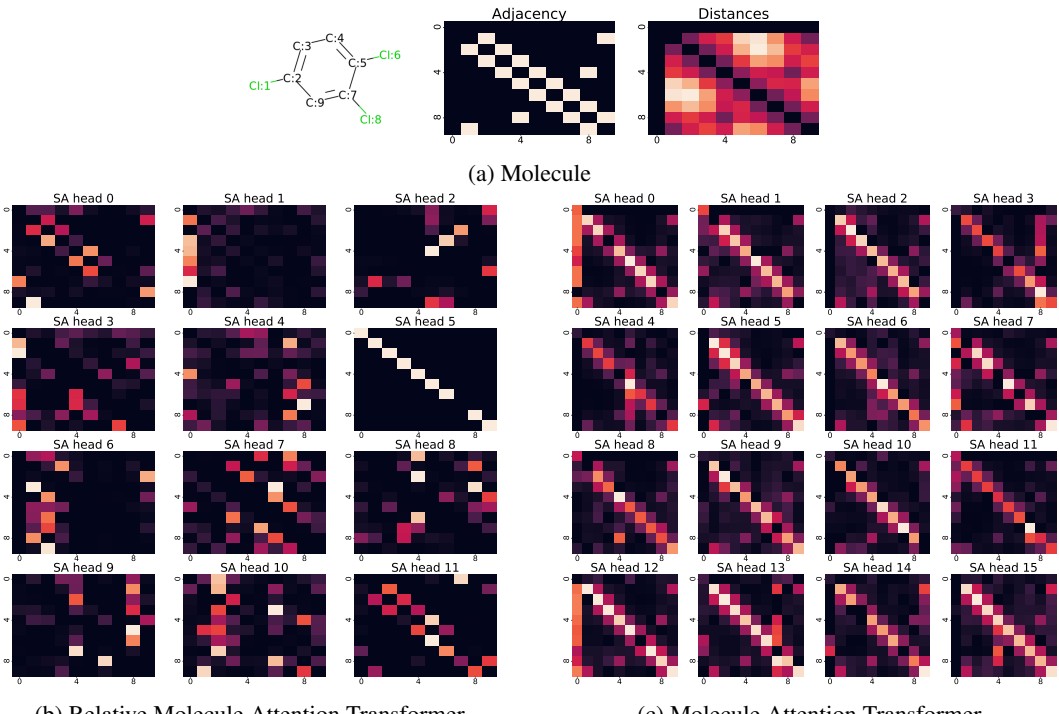

(a) Molecule

(b) Relative Molecule Attention Transformer

(c) Molecule Attention Transformer

Figure 8: Visualization of the learned self-attention for each of all attention heads in the second layer of pretrained R-MAT (left) and all attention heads in pretrained MAT (right), for a molecule from the ESOL dataset. The top Figure visualizes the molecule and its adjacency and distance matrices. The self-attention pattern in MAT is dominated by the adjacency and distance matrix, while R-MAT seems capable of learning more complex attention patterns.

informed our choices. We also hope it will inform future efforts in designing attention mechanism for molecular data. We follow here the same evaluation protocol as in Section 4.4 and show how different natural variants compare against R-MAT.

### E.1 SELF-ATTENTION VARIANTS

Relative Molecule Self-Attention is designed to better incorporate the relative spatial position of atoms in the molecule. The first step is embedding each pair of atoms. Then, the embedding is used to re-weight self-attention. To achieve this, Relative Molecule Self-Attention combines ideas from natural language processing (Shaw et al., 2018; Dai et al., 2019; Huang et al., 2020). These works focus on encoding better relative positions of tokens in the input.

We compare to three specific variants from these works that can be written using our previously introduced notation as:

1. Relative self attention (Shaw et al., 2018):
$$e_{ij} = (x_i W^Q)(x_j W^K)^T + (x_i W^Q)\boldsymbol{b}_{ij}^K.$$

2. Relative self attention with attentive bias (Dai et al., 2019):

$$e_{ij} = (x_i W^Q)(x_j W^K)^T + (x_i W^Q)\boldsymbol{b}_{ij}^K + \\ + \boldsymbol{u}^T(x_j W^K) + \boldsymbol{v}^T \boldsymbol{b}_{ij}^K.$$

3. Improved relative self-attention (Huang et al., 2020):
$$e_{ij} = (x_i W^Q)(x_j W^K)^T + (x_i W^Q)\boldsymbol{b}_{ij}^K + \\ + (x_j W^K)\boldsymbol{b}_{ij}^K.$$

Table 11 shows that the attention operation used in R-MAT outperforms variants 2 and 3 across all three tasks and variant 1 on two tasks, being comparable on the third one. This might be expected given that Relative Molecule Self-Attention combines these ideas (c.f. Equation (3)).

Table 11: Test set performances of R-MAT for different choices of the relative self-attention.

|  | BBBP | ESOL | FreeSolv |
|---|---|---|---|
| R-MAT | $.908_{(.039)}$ | $.378_{(.027)}$ | $.438_{(.036)}$ |
| Relative type = 1 | $.859_{(.057)}$ | $.371_{(.041)}$ | $.509_{(.028)}$ |
| Relative type = 2 | $.856_{(.049)}$ | $.424_{(.014)}$ | $.472_{(.057)}$ |
| Relative type = 3 | $.882_{(.051)}$ | $.389_{(.040)}$ | $.441_{(.021)}$ |

### E.2 Enriching bond features with atom features

In Relative Molecule Self-Attention, we use a small number of bond features to construct the atom pair embedding. We investigate here the effect of extending bond featurization.

Inspired by (Shang et al., 2018), we added information about the atoms that an edge connects. We tried three different variants. In the first one, we extend the bond representation with concatenated input features of atoms that the bond connects. In the second one, instead of raw atoms' features, we tried the one-hot-encoding of the type of the bond connection (i.e. when the bond connects atoms C and N, we encode it as a bond 'C_N' and take the one-hot-encoding of this information). Finally, we combined these two approaches together.

Table 12: Test set performances of R-MAT for different choices of bond featurization.

|  | BBBP | ESOL | FreeSolv |
|---|---|---|---|
| R-MAT | $.908_{(.039)}$ | $.378_{(.027)}$ | $.438_{(.036)}$ |
| Connected atoms features | $.866_{(.073)}$ | $.406_{(.048)}$ | $.489_{(.046)}$ |
| Connection type one-hot | $.863_{(.012)}$ | $.411_{(.028)}$ | $.510_{(.055)}$ |
| Both | $.873_{(.034)}$ | $.390_{(.020)}$ | $.502_{(.044)}$ |

The results are shown in Table 12. Surprisingly, we find that adding this type of information to the bond features negatively affects performance of R-MAT. This suggests that R-MAT can already access these features efficiently from the input (which we featurize using the same set of features). This could also happen due to the fact that after a few layers, the attention is not calculated over the input atoms anymore. Instead, it works over hidden embeddings, which themselves can be mixed representations of multiple atom embeddings (Brunner et al., 2019), where the proposed additional representation contains only information about the input features.

### E.3 Distance encoding variants

R-MAT uses a specific radial base distance encoding proposed by (Klicpera et al., 2020), followed by the envelope function, with $N_{emb} = 32$. We compare here to several other natural choices.

We tested the following distance encoding variants : (1) removal of the envelope function, (2) increasing the number of distance radial functions to 128, (3) using distance embedding from the popular MAT model (Maziarka et al., 2020), (4) using distance embedding from the popular SchNet model (Schütt et al., 2017). The distance in MAT is encoded as $e(d) = \exp(-d)$. The distance in SchNet is encoded as $e_n(d) = \exp(-\gamma \|d - \mu_n\|^2)$, for $\gamma = 10$Å and $0$Å $\leq \mu_n \leq 30$Å divided into $N_{emb}$ equal sections, with $N_{emb}$ set to 32 or 128.

The results are shown in Table 13. These results corroborate that a proper representation of distance information is a key in adapting self-attention to molecular data. We observe that all variants underperform compared to the radial base encoding used in Relative Molecule Self-Attention. Noteworthy

is the fact that encoding distances in MAT way caused a lot of instability during the training of R-MAT.

Table 13: Test set performances of R-MAT for different choices of distance modeling.

|  | BBBP | ESOL | FreeSolv |
|---|---|---|---|
| R-MAT | $.908_{(.039)}$ | $.378_{(.027)}$ | $.438_{(.036)}$ |
| $N_{emb} = 128$ | $.850_{(.102)}$ | $.417_{(.025)}$ | $.427_{(.016)}$ |
| no envelope | $.887_{(.025)}$ | $.397_{(.047)}$ | $.473_{(.019)}$ |
| $N_{emb} = 128$, no envelope | $.901_{(.030)}$ | $.416_{(.014)}$ | $.452_{(.008)}$ |
| MAT dist | $.886_{(.024)}$ | $.404_{(.045)}$ | $.444_{(.012)}$ |
| SchNet dist | $.883_{(.065)}$ | $.398_{(.043)}$ | $.490_{(.033)}$ |
| $N_{emb} = 128$, SchNet dist | $.888_{(.054)}$ | $.400_{(.043)}$ | $.445_{(.010)}$ |

# F   ADDITIONAL COMPARISON OF GRAPH PRETRAINING

## F.1   PRETRAINING METHODS BENCHMARK

As pretraining is nowadays the main component of big Transformer architectures (Devlin et al., 2018; Liu et al., 2019; Clark et al., 2020), we decided to devote more attention to this issue. For this purpose we compared various graph pretraining methods to identify the best one and use it in the final R-MAT model.

**Pretraining methods**   We used various pretraining methods proposed in the molecular property prediction literature (Hu et al., 2020; Maziarka et al., 2020; Rong et al., 2020; Fabian et al., 2020). To be more specific, we tried R-MAT with the following pretraining procedures:

- No pretraining – as a baseline we include reuslts for R-MAT fine-tuned from scratch, without any pretraining.
- Masking – masked pretraining used in (Maziarka et al., 2020). This is an adaptation of standard MLM pretraining used in NLP (Devlin et al., 2018) to the graphical data. In this approach, we mask features of 15% of all atoms in the molecule and then pass it throught the model. The goal is to predict what was the masked features.
- Contextual – contextual pretraining method proposed by (Rong et al., 2020). We described it further in Appendix B.
- Graph-motifs – graph-level motif prediction method proposed by (Rong et al., 2020), where for every molecule we obtain the fingerprint with information whether specified molecular functional groups are present in our molecule. The network's task is multi-label classification, where it has to predict, whether every predefined functional group is in the given molecule.
- Physicochemical – graph-level prediction method proposed by (Fabian et al., 2020). We described it further in Appendix B.
- GROVER – pretraining used by authors of GROVER (Rong et al., 2020). Combination of two pretraining methods: contextual and graph-motifs.
- R-MAT– pretraining used in this paper. Combination of two pretraining methods: contextual and physicochemical.

**Experimental setting**   We pretrained every model using described earlier dataset with 4M molecules. For every pretraining choice, we trained the model for 50 epochs, using the same training settings as for our standard pretraining (see Appendix C.3).

Model hyperparameters and fine-tuning settings were the same as for MAT and GROVER benchmarks. We used the BBBP, ESOL and FreeSolv datasets, splitted using a scaffold split, with 3 different data splits. Data splits are different than in previous experiments, and this causes the fact that the results are differs from the results from other paragraphs'.

**Results** Results of this benchmark are presented in Table 14. We can draw some interesting conclusions from them. One can see, that using any kind of pretraining helps in obtaining better results than for the model trained from scratch. Using physicochemical features for graph-level training gives better results than graph-motifs. Therefore R-MAT pretraining (contextual + physicochemical) is better than GROVER pretraining (contextual + graph-motfis). Moreover combination of two tasks in pretraining usually gives better results than pretraining using only one task. Interestingly, both node-level pretraining methods (masking and contextual pretraining), returns similar results.

Table 14: Test set performances of R-MAT for different choices of pretraining used.

|  | BBBP | ESOL | FreeSolv |
|---|---|---|---|
| No pretraining | $.855_{(.081)}$ | $.423_{(.021)}$ | $.495_{(.016)}$ |
| Masking | $.867_{(.046)}$ | $.377_{(.016)}$ | $.407_{(.074)}$ |
| Contextual | $.901_{(.039)}$ | $.382_{(.034)}$ | $.413_{(.047)}$ |
| Graph-motifs | $.876_{(.035)}$ | $.389_{(.041)}$ | $.473_{(.092)}$ |
| Physiochemical | $.897_{(.042)}$ | $.406_{(.072)}$ | $.400_{(.085)}$ |
| GROVER | $.897_{(.022)}$ | $.378_{(.027)}$ | $.455_{(.062)}$ |
| R-MAT | $.893_{(.045)}$ | $.360_{(.012)}$ | $.402_{(.029)}$ |

### F.2 PRETRAINING LEARNING CURVES

In Figure 9 we present learning curves for R-MAT pretrained with our procedure (contextual + physiochemical). Left-side image shows that training loss flattens out quite quickly, however it slowly decreases until the end of the training. Moreover one can see that contextual task is harder for the network than graph-level property prediction, as their losses vary by several orders of magnitude. Left and middle images shows that R-MAT predictions for validation datasets are of good quality, moreover these curves also presents that our model learns all the time, reaching the best values at the end of training

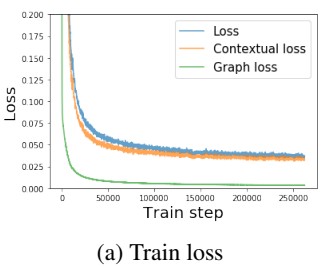 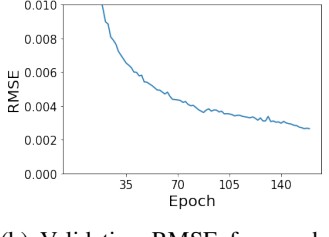 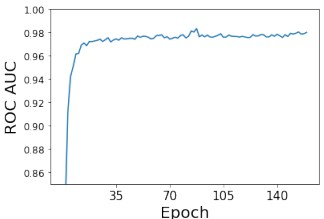

(a) Train loss

(b) Validation RMSE for graph-level property prediction task

(c) Validation ROC AUC for contextual task

Figure 9: Learning curves for R-MAT pretraining. On the left-side figure one can see train losses for both contextual and graph-level property prediction tasks. On the middle figure one can see the RMSE for graph-level property prediction for validation dataset obtained by R-MAT during pretraining. On the right-side figure one can see ROC AUC for the classification task of contextual prediction for validation dataset obtained by R-MAT during pretraining.

We also tested, whether longer pretraining allows R-MAT to get better results during the fine-tuning process. In Figure 10 we present fine-tuning scores obtained by models pretrained with different number of pretraining epochs, for FreeSolv and ESOL datasets. Interestingly, longer training does not always results in better fine-tuning scores. This is indeed the case for the FreeSolv dataset (left-side image), however for ESOL (right-side image) we cannot draw such a conclusion.

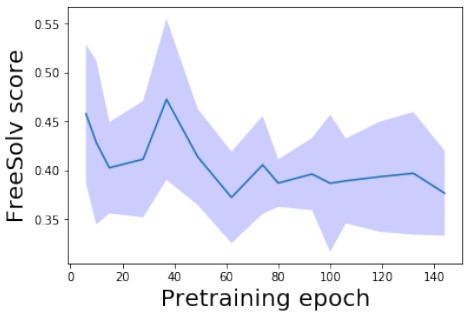

(a) FreeSolv score for R-MAT pretrained for given numbers of pretraining epochs

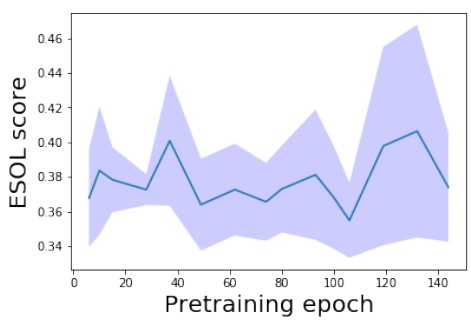

(b) ESOL score for R-MAT pretrained for given numbers of pretraining epochs

Figure 10: Fine-tuning scores obtained by R-MAT pretrained with a different number of pretraining epochs.

