# OpenReview forum: "Relative Molecule Self-Attention Transformer"
_ICLR.cc/2022/Conference — ICLR 2022 Submitted_

### Official Review · Reviewer_DtcY · 2021-11-01

**Correctness:** 3
**Technical Novelty And Significance:** 2
**Empirical Novelty And Significance:** 2
**Recommendation:** 6
**Confidence:** 3

**Main Review:**

Strengths:

The proposed relative self-attention is carefully designed to consider three factors: relative distance, shortest path distance in the molecular graph, and physiochemical. Inspired by the positional bias and content bias from NLP methods, a novel relative molecule self-attention is proposed and helps the Transformer model achieve good experimental performance.

Weaknesses:

Some concerns about the evaluation. In Table 1, why do only two datasets (BBBP and Estrogen-$\beta$) use scaffold splits, and the other four datasets use the random split? The baseline method GROVER evaluates on multiple graph classification datasets, why only BBBP is evaluated in Table 1? While another dataset BACE is only evaluated in Table 2? Could the authors include results on more datasets like GROVER? Evaluating only on selected datasets makes the readers suspicious of the generalization of the proposed method.



**Summary Of The Paper:**

This paper proposes a relative self-attention layer for the Transformer model. The relative self-attention of two atoms consists of their relative distance, their shortest path distance in the molecular graph, and their physiochemical. The proposed relative molecule attention Transformer can be first pretrained with a contextual property prediction task and then a graph-level prediction task. The pretrained Transformer can be finetuned on downstream molecular property prediction tasks and achieves excellent performance.

**Summary Of The Review:**

The paper proposes a novel relative molecule self-attention layer for the Transformer model. The model can be first pretrained and then finetuned on downstream molecule property prediction tasks. Experimental results demonstrate the effectiveness of the proposed method. However, there are some concerns about the evaluation as described in the Weaknesses section.

---

> ### Author Response · Authors · 2021-11-19
> **Response to Reviewer DtcY**
>
> Thank you very much for your review and your time.
>
> > Weaknesses:
>
> > Some concerns about the evaluation. In Table 1, why do only two datasets (BBBP and Estrogen-beta) use scaffold splits, and the other four datasets use the random split?
>
> We decided to directly compare on benchmarks from our two main competing methods (MAT [1] and GROVER [2]), despite their shortcoming or added inconsistency. We wanted to compare R-MAT to other models on the two existing benchmarks, one proposed by Maziarka et al [1] and another proposed by Rong et al [2]. Thus we decided to use datasets and splits given by these authors. We have clarified this in the text. We are sorry that this is confusing -- it is a tradeoff between having inconsistent evaluation setup and making it more convincing that we improve upon previous results (i.e. that we do not adapt benchmarks to our method).
>
> > The baseline method GROVER evaluates on multiple graph classification datasets, why only BBBP is evaluated in Table 1? While another dataset BACE is only evaluated in Table 2? Could the authors include results on more datasets like GROVER? Evaluating only on selected datasets makes the readers suspicious of the generalization of the proposed method.
>
> We wanted to add all single task datasets from the benchmark in GROVER [2] publication, because they form 6 diverse datasets, however due to some technical problems, we delivered only 4. We have added results for additional 2 datasets in the updated version of manuscript. In the process, we also found a bug in processing results on the Lipo dataset, where on the large-scale grid GROVER now outperforms R-MAT. On the whole, R-MAT achieves a similar performance to GROVER in Table 2.
>
> [1] Łukasz Maziarka, Tomasz Danel, Sławomir Mucha, Krzysztof Rataj, Jacek Tabor, Stanisław Jastrzębski, Molecule Attention Transformer, 2020.
> [2] Yu Rong, Yatao Bian, Tingyang Xu, Weiyang Xie, Ying Wei, Wenbing Huang, Junzhou Huang, Self-Supervised Graph Transformer on Large-Scale Molecular Data, 2020.

---

> > ### Comment · Reviewer_DtcY · 2021-11-29
> > **Author response sovled my conern.**
> >
> >  Thank the authors' response, which solved my concerns. After all the review and author responses, I decided to maintain my score.

---

> > > ### Author Response · Authors · 2021-11-29
> > > **Thank you**
> > >
> > > Thank you very much for reading the rebuttal and supporting the paper.

---

### Official Review · Reviewer_GZ33 · 2021-11-02

**Correctness:** 4
**Technical Novelty And Significance:** 2
**Empirical Novelty And Significance:** 3
**Recommendation:** 6
**Confidence:** 2

**Main Review:**

- Pros
    - The experimental results are comprehensive.
    - The paper is easy to follow. Well written.

- Cons
    - Lack of intuitive understanding of proposed features w.r.t. tasks to solve.
    - Theoretical justification on why it is impossible to learn these features with the vanilla Transformer or how difficult to learn such features.
    - Difficult to reproduce the result due to the large scale experiments

- Questions
    - How many overlaps between the task datasets and pre-trained datasets? Does this overlap influence the test performance?
    - Increasing the maximum neighborhood order does not improve the performance.
        - The results seem inconsistent. How do we interpret the result?
    - What makes EGNN perform the best with the QM9 dataset (Figure 3) and why the proposed method cannot achieve a similar result?

**Summary Of The Paper:**

The paper proposes a new transformer network architecture to pre-train the molecule datasets. Based on the Molecule Attention Transformer, the proposed model, R-MAT, incorporates a few handcrafted features into the self-attention layer of a transformer architecture. The features can incorporate distances between atoms from multiple perspectives. The experimental results show that the pre-trained model is useful to predict various properties of molecules.

**Summary Of The Review:**

Based on the pros and cons written above, I think the paper is marginally above the acceptance threshold.

---

> ### Author Response · Authors · 2021-11-19
> **Response to Reviewer GZ33**
>
> Thank you very much for your review and your time.
> >Cons
>
> > Lack of intuitive understanding of proposed features w.r.t. tasks to solve.
>
> We agree the lack of interpretability is a challenge. It is quite clear that R-MAT outperforms MAT by a significant margins on tasks were methods that are more directly encoding the 3d dimensional structure of the molecule generally (in the literature) outperform ones that do not (such as FreeSolv, ESOL, and QM7). Therefore, we can take from this that distance information is important for these tasks and is efficiently encoded by R-MAT. This is further supported by ablation studies and qualitative results in Figure 4.
> To further stress the importance of distance representation, we added additional ablation in Table 11 in Appendix E.3
>
> > Theoretical justification on why it is impossible to learn these features with the vanilla Transformer or how difficult to learn such features.
>
> That’s a very good point. We think we first should split features into ones that increase expressiveness of the model (or more precisely, make it significantly more easily to express given functions, model is able to represent any function given sufficient size), and ones that do not (or more precisely, it is not clear what function they make easier to learn).
> Echoing our answer to Reviewer 1, there is a good argument for why one might need more flexibility in representing distance information. If we were to represent distances as in MAT (i.e. scalar time distance), it would severely limit expressivity of the model. In such a scenario, the attention strength between a pair of atoms has to depend monotonically on the distance. More theoretical arguments can be found also in [1,2] who introduced similar distance encoding to the one we are using.
> Another feature that makes it much easier to learn something is relative encoding. Following remarks in papers that introduced relative encoding, it makes it significantly easier for the model to represent that attention strength should be calculated the same way for two pairs of molecules even if they are in two very different parts of the molecule.
> We also use rdkit features such as using atom and bond features when computing attention strength. These are surely learnable from data, and it is an interesting challenge for the future to reduce our reliance on them. Likely, it is related to pretraining tasks that are too simple to force the model to learn such features.
> We will add this discussion to the paper.
>
> > Difficult to reproduce the result due to the large scale experiments
>
> We very much share your sentiment that it is key to lower the barrier for reproducing results. This is precisely the reason we focus on experiments on showing that R-MAT can be tuned to a very competitive performance with just tuning the learning rate. We also will open-source our code and pretrained weights. The few large-scale experiments included should be seen as additional corroboration of the fact that the model achieves SOTA or very competitive performance.
>
> > Questions
>
> > How many overlaps between the task datasets and pre-trained datasets? Does this overlap influence the test performance?
>
> We added the information about overlap of pre-trained dataset with all tasks datasets in Table 8 in Appendix C.3. Usually at most few percent of molecules from tasks datasets appear in 4M pre-training smiles. However, overall molecules used for pre-training are similar to those used in tasks, as for pre-training we used a big drug-like subset of ZINC and ChEMBL datasets. Nonetheless this does not affect the test performance, as pre-training consists of semi-supervised task of context prediction and other task associated with the prediction of simple physico-chemical characteristics, loosely related to the fine-tuning tasks, which are additionally added to the final graph embedding during the prediction.
>
> > Increasing the maximum neighborhood order does not improve the performance. The results seem inconsistent. How do we interpret the result?
>
> We interpret this as reflecting the chemical reality that atoms that are very far away are less likely to influence each other's physical properties (because they can have widely different spatial arrangement once they are many hops away). We will add a clarification in the paper.
>
> > What makes EGNN perform the best with the QM9 dataset (Figure 3) and why the proposed method cannot achieve a similar result?
>
> We were able to improve the results but did not outperform EGNN. This is surprising given that the size of the datasets included in QM9 is large enough so that the weaker inductive bias of our model shouldn’t be an issue (echoing similar results when comparing Vision Transformer and ResNet on large datasets).
>
>
> [1] Kristof T. Schütt et al., SchNet: A continuous-filter convolutional neural network for modeling quantum interactions, 2017.
>
> [2] Johannes Klicpera et al., Directional Message Passing for Molecular Graphs, 2020.

---

### Official Review · Reviewer_Lgqf · 2021-11-02

**Correctness:** 3
**Technical Novelty And Significance:** 2
**Empirical Novelty And Significance:** 3
**Recommendation:** 6
**Confidence:** 4

**Main Review:**

Adding distance information directly to the learned "distance" before computing the softmax as well as additional information as in MAT is a very good idea (similar to ALiBi [1]). Results in Table 3 are supportive for adding distance-, bond- and a neighborhood-embedding in the attention matrix.
There are datasets the support a performace increase of the proposed method over baselines and established methods, however
 1. Distance Matrix in MAT vs. In R-MAT: is that it is first embedded, then a learn projection; rather than taking the distance matrix directly times an e.g. a learned scalar; is there a clear advantage of one over the other?
 2. Table 1 reports the best performance for ESOL and FreeSolv, given that R-MAT was pretrained on 200 rdkit-descriptors and in one model-variant uses those descriptors in the last layer, please make sure to also include a baseline variant of e.g. RF with the 200 rdkit-descriptors (a few of whom describe solubility), (additional to using fingerprints)
 3. Please state why between the Experiments (1-2) splits were changed from random to scaffold, as well as the normalization of the labels
 4. In Table 9: "R-MAT outperforms other variants across the three tasks" however in ESOL type 1 performs best/not significantly differently
 5. One disadvantage is the costly calculation of the distance matrix, which should be mentioned at least once in the paper
 6. Reproducibility: Code to reproduce the results is not provided


Minor Issues:
- "we report the mean test score" but the values in brackets for table 1, 2, .., are they std, var, or SD?
- How were the fixed-hyperparameters for Experiment 1, small-hyperparameter budget selected?
- 8 nVidia A100 GPUs --> Nvidia
- Table 2: QM7 should be changed to QM9
- Equation (1): by adding to vanilla self-attention e_{ij} , doesn't have a variance of sqrt(d_z), so the denominator in the softmax might be adjusted
- The distance matrix might change dependent on calculation, any improvements from running multiple runs?; distance is not static; rotatable bonds

[1] Press, O., Smith, N.A., & Lewis, M. (2021). Train Short, Test Long: Attention with Linear Biases Enables Input Length Extrapolation. ArXiv, abs/2108.12409.

**Summary Of The Paper:**

The paper introduces Relative Molecule Attention Transformer, adding distance-, bond- and a neighborhood-embedding  in the attention matrix, and pretrain the model on 4M molecules via context pretraining and on 200 rdkit-descriptors.

**Summary Of The Review:**

Several issues have been discussed: Experimental results should be compared to additional baselines, changes in the experimental setup are not justified and seem unnecessary, drawbacks are not adequately discussed.  In the current form, the paper is marginally below the acceptance threshold but the reviewer is inclined to change the rating, in light of new supportive evidence.

**update**: The authors successfully tackled the issues I raised with the manuscript, therefore I am happy to raise my score.

---

> ### Author Response · Authors · 2021-11-19
> **Response to Reviewer Lgqf (part 2)**
>
> > Minor Issues:
>
> > "we report the mean test score" but the values in brackets for table 1, 2, .., are they std, var, or SD?
>
> This is the standard deviation across all splits. We have clarified this in the text.
>
> > How were the fixed-hyperparameters for Experiment 1, small-hyperparameter budget selected?
>
> We chose them manually by taking the extreme values defining the big grid search, and narrowing them as much as possible without hurting the performance.
>
> > 8 nVidia A100 GPUs --> Nvidia
>
> Corrected.
>
> > Table 2: QM7 should be changed to QM9
>
> Actually this is the score for the QM7 dataset that was used in the original GROVER benchmark [4].
>
> > Equation (1): by adding to vanilla self-attention e_{ij} , doesn't have a variance of sqrt(d_z), so the denominator in the softmax might be adjusted
>
> That’s a very good point. Indeed the variance is larger than sqrt(d_z). Following other similar works [6,7], we left the partial normalization, but it might be worth investigating in more detail in the future.
>
> > The distance matrix might change dependent on calculation, any improvements from running multiple runs?; distance is not static; rotatable bonds
>
> We tried to add multiple conformations in train as well as in test time, however the results were pretty similar to the model trained using only one conformation. Although we think that this is a nice idea for future work, which could be especially useful for datasets such as QM9 or tasks such as docking
>
> [1] Kristof T. Schütt, Pieter-Jan Kindermans, Huziel E. Sauceda, Stefan Chmiela, Alexandre Tkatchenko, Klaus-Robert Müller, SchNet: A continuous-filter convolutional neural network for modeling quantum interactions, 2017.
>
> [2] Johannes Klicpera, Janek Groß, Stephan Günnemann, Directional Message Passing for Molecular Graphs, 2020.
>
> [3] Łukasz Maziarka, Tomasz Danel, Sławomir Mucha, Krzysztof Rataj, Jacek Tabor, Stanisław Jastrzębski, Molecule Attention Transformer, 2020.
>
> [4] Yu Rong, Yatao Bian, Tingyang Xu, Weiyang Xie, Ying Wei, Wenbing Huang, Junzhou Huang, Self-Supervised Graph Transformer on Large-Scale Molecular Data, 2020.
>
> [5] Press, O., Smith, N.A., & Lewis, M. (2021). Train Short, Test Long: Attention with Linear Biases Enables Input Length Extrapolation. ArXiv, abs/2108.12409.
>
> [6] Peter Shaw, Jakob Uszkoreit, Ashish Vaswani, Self-Attention with Relative Position Representations, 2018.
>
> [7] Zhiheng Huang, Davis Liang, Peng Xu, Bing Xiang, Improve Transformer Models with Better Relative Position Embeddings, 2020.

---

> ### Author Response · Authors · 2021-11-19
> **Response to Reviewer Lgqf**
>
> Thank you very much for your review and your time.
>
> > 1. Distance Matrix in MAT vs. In R-MAT: is that it is first embedded, then a learn projection; rather than taking the distance matrix directly times an e.g. a learned scalar; is there a clear advantage of one over the other?
>
> That’s a very good point. Indeed a natural approach is to take the distance matrix multiplied by a scalar.
> However, it limits the expressivity of the model. R-MAT can much more easily express functions such as “relate these two atoms if they are further than 2A but closer than 4A”. Similar view is presented in the papers that introduced the RBF envelope that we use to encode the distance information [1,2].
> We support this view both qualitatively and quantitatively.
> Qualitatively, we show this in Figure 4. Visibly, R-MAT learns much more complex attention patterns.
> Quantitatively, we show extensive ablation studies in the Appendix. We have also added a new ablation, with MAT-related distance encoding to Table 11 in Appendix E.3. Please let us know if you think it makes more sense to move these results to the main text.
>
> > 2. Table 1 reports the best performance for ESOL and FreeSolv, given that R-MAT was pretrained on 200 rdkit-descriptors and in one model-variant uses those descriptors in the last layer, please make sure to also include a baseline variant of e.g. RF with the 200 rdkit-descriptors (a few of whom describe solubility), (additional to using fingerprints)
>
> Thank you for this great suggestion. Indeed, RF with rdkit features is a much stronger baseline. Interestingly, R-MAT is the only model that overall improves upon it (losing on two tasks in Table 1). Given the relevance of the baseline, we also added it to Table 2, where R-MAT wins with it consistently.
>
> > 3. Please state why between the Experiments (1-2) splits were changed from random to scaffold, as well as the normalization of the labels
>
> We decided to directly compare on benchmarks from our two main competing methods (MAT [3] and GROVER [4]), despite their shortcoming or added inconsistency.. We have clarified this in the text. We are sorry that this is confusing -- it is a tradeoff between having inconsistent evaluation setup and making it more convincing that we improve upon previous results (i.e. that we do not adapt benchmarks to our method).
>
> > 4. In Table 9: "R-MAT outperforms other variants across the three tasks" however in ESOL type 1 performs best/not significantly differently
>
> Thank you for drawing our attention to this. We have changed the text. Overall R-MAT outperforms other variants.
>
> > 5. One disadvantage is the costly calculation of the distance matrix, which should be mentioned at least once in the paper
>
> We fully agree.  We added a clarification and calculation time for FreeSolv, ESOL and BBBP datasets in Appendix C.2, Table 7 and Figure 5.
> We used a simple and quite fast algorithm. It takes on average 8 ms per molecule for FreeSolv (datasets with smallest molecules), 21 ms per molecule for ESOL (datasets with medium-sized molecules) and 131 ms per molecule for BBBP (datasets with biggest molecules). Exploring other methods (more accurate or faster) is an interesting topic for the future. On QM9 we used the provided distance matrices.
> We also think it is likely that the model could be trained to infer the distance information by, for example, removing it at random during training. We leave this for the future.
>
> > 6. Reproducibility: Code to reproduce the results is not provided
>
> Sorry, we have ready code for release but didn’t make the final push to include it in the supplement. Please find the code of the R-MAT model in the supplement. Furthermore, we will release the pre-trained weights.

---

> ### Author Response · Authors · 2021-11-29
> **Thank you**
>
> Thank you very much for taking the time to engage with our rebuttal and raising the score!

---

### Author Response · Authors · 2021-11-19
**To all Reviewers**

We thank all the reviewers for their time.

Here are the highlights of changes we introduced to address the comments (in blue in the revised version):
- Reviewer 1 suggested adding RF with more rdkit features as a baseline to some of the experiments (Table 1 and Table 2). In Table 1 experiments R-MAT turns out to be the only model that outperforms the baseline. In particular, the baseline model outperforms MAT and GROVER. This is interesting, especially that GROVER depended on these additional rdkit features (MAT did not).
- We added an ablation study showing the fundamental importance of the proposed distance encoding. This helps address both Reviewer 1 and Reviewer 2 comments about the utility of distance information.
-Reviewer 2 and 3 raised issues about differences in benchmarks between Table 1 and Table 2. This was a necessary tradeoff between having convincing results (i.e. that we did not adapt benchmarks to our method), and having inconsistent evaluation schemes between the two Tables. We are sorry that this caused confusion. We have added clarification to the paper.
- Reviewer 3 raised concern about including only 4 benchmarks from the GROVER paper. We have added the remaining two single-task datasets. In the process, we also found a bug in processing results on the Lipo dataset, where on the large-scale grid GROVER now outperforms R-MAT. On the whole, R-MAT achieves a similar performance to GROVER in Table 2.
- We have uploaded code as part of the supplement.

---

### Decision · Program_Chairs · 2022-01-20

**Decision:**

Reject

**Comment:**

This paper presents an extension to the MAT model by using relative attention that considers graph-level distance, geometric distance, and bond type between nodes during the attention computation.  This is shown to lead to improved performance on several benchmark datasets against MAT and GROVER.  The inclusion of additional information into the attention computation is a sensible and natural choice for transformer with the success of relative positional embedding in the NLP and vision domains. But it is somewhat a straightforward extension of the existing ideas from other domains to MAT, hence the novelty is somewhat limited.  It is also worth noting that the proposed method should be considered in the context of the larger body of research on 3D GNNs. The authors drew inspiration from DimeNet's design in the encoding of geometric distances but do not consider it in the empirical comparisons. Instead, it focused exclusively on transformer based models. This limits the scope of conclusions that we can draw from these experiments and makes it difficult to gauge the practical impact of RMAT in comparison to many other GNN methods that uses 3D geometries of the molecule.